# Inflammation and Oxidative Damage in Ischaemic Renal Disease

**DOI:** 10.3390/antiox10060845

**Published:** 2021-05-25

**Authors:** Áine M. de Bhailís, Constantina Chrysochou, Philip A. Kalra

**Affiliations:** Department of Nephrology, Salford Royal NHS Foundation Trust, Salford M6 8HD, UK; tina.chrysochou@srft.nhs.uk (C.C.); philip.kalra@srft.nhs.uk (P.A.K.)

**Keywords:** oxidative stress, ischaemic renal disease, atherosclerotic renovascular disease, renal inflammation

## Abstract

Ischaemic renal disease as result of atherosclerotic renovascular disease activates a complex biological response that ultimately leads to fibrosis and chronic kidney disease. Large randomised control trials have shown that renal revascularisation in patients with atherosclerotic renal artery disease does not confer any additional benefit to medical therapy alone. This is likely related to the activation of complex pathways of oxidative stress, inflammatory cytokines and fibrosis due to atherosclerotic disease and hypoxic injury due to reduced renal blood flow. New evidence from pre-clinical trials now indicates a role for specific targeted therapeutic interventions to counteract this complex pathogenesis. This evidence now suggests that the focus for those with atherosclerotic renovascular disease should be a combination of revascularisation and renoprotective therapies that target the renal tissue response to ischaemia, reduce the inflammatory infiltrate and prevent or reduce the fibrosis.

## 1. Introduction

Reactive oxygen species are produced as part of normal cellular metabolism and as a result of environmental factors such as pollutants. They are highly reactive and can damage cell structures. Such oxidative damage results from an excess of oxidant radicals and because of insufficient degradation of these by antioxidant systems that serve as defence mechanisms [1]. Oxidative damage is clinically important and has been shown to play a significant role in many pathological conditions, including malignancy, atherosclerosis, hypertension, ischaemic heart disease and chronic kidney disease (CKD).

The kidneys are actually extremely metabolically active organs as they receive 25% of the total cardiac output and consume 7% of daily energy expenditure to support their various functions. They are in turn highly susceptible to oxidative damage. The mitochondrial respiratory chain and nicotinamide adenine dinucleotide phosphate(NADPH) oxidases are the main source of free radicals in the kidney, whereas its defences include antioxidant enzymes such as superoxide dismutase and catalase [2]. Previous studies have highlighted that patients with end stage kidney disease (ESKD) have much lower rates of antioxidant activity [3].

Ischaemic kidney disease, also referred to as ischaemic nephropathy or atherosclerotic renovascular disease (ARVD), is a form of CKD in which oxidative stress plays a pre-eminent role in pathogenesis. In this review, we will consider the following aspects of ischaemic renal disease, with evidence deriving from both pre-clinical studies and from patients with ARVD:The related pathophysiological processes of inflammation, oxidative stress and fibrosis leading to tissue injury;Biomarker evidence for the importance of oxidative injury;The application of imaging techniques to determine intra-renal ischaemia;Emerging therapeutic strategies to counteract this complex pathogenesis.

## 2. Materials and Methods

### Review Methodology

A literature search was carried out that included the published literature from January 2010 to January 2020 that focused on the pathophysiology of atherosclerotic renovasular disease. Information was retrieved from PubMed through the National Library of Medicine using keywords such as “atherosclerotic renovascular disease”, “oxidative stress and ischaemic nephropathy” and “inflammation and ischaemic nephropathy”. The search yielded 612 papers, including randomized control trials, original articles, review articles, clinical trials and original research. These articles were evaluated based on how relevant the article was and if the article addressed a research question related to the pathophysiology and treatment of ischaemic nephropathy. Articles were excluded if they were deemed irrelevant to the purposes of this review. Sixty-six of these were considered relevant to the topic. Other manuscripts were located through reference lists of relevant articles. Some articles were included that were published before 2010 as they had established important concepts and knowledge that were relevant to the topic.

## 3. Results

### 3.1. Ischaemic Nephropathy

Atherosclerotic renal artery disease is a common condition that increases with age and is known to affect 7% of those aged >65 years in the community [4]. When associated with renal impairment, it is termed ischaemic nephropathy [4]. ARVD is often associated with atheromatous disease in other vascular systems, and patients have an increased mortality [5]; it remains an important cause of ESKD based on large registry data such as the USRDS data [6].

In previous decades, there was a huge interest in revascularisation as a potential cure for both the associated renal impairment and renovascular hypertension with a large increase in the number of revascularisation procedures [7]. However, large randomised control trials such as STAR in 2009 [8], ASTRAL in 2009 [9] and CORAL in 2014 [10] have effectively shown little overall clinical benefit following revascularisation of unselected patients with haemodynamically significant lesions (Table 1). This has led to a dramatic drop in the number of revascularisation procedures, which are now reserved for complex cases such as those with acute heart failure syndromes, rapidly progressive renal impairment or hypertensive emergencies unresponsive to medical management [11].

Previous animal studies had suggested that the main insult resulting from renal artery stenosis (RAS) was a haemodynamic one as a result of progressive occlusion of the renal vasculature leading to hypoperfusion and ultimately ischaemic kidney disease. Studies in porcine models have shown that renal perfusion does not decrease until there is a reduction of >42% in the lumen diameter with a dramatic drop once the RAS >70% of the diameter. Concordantly, plasma renin activity starts to increase once there is a reduction of >58% of the diameter. Interestingly, a threshold RAS diameter for an increase in mean arterial pressure (MAP) was not identifiable [12], which suggests that the effects of RAS are not limited to a haemodynamic effect.

In recent years, there has been a shift in the understanding of the disease and recognition that in many cases there is a large inflammatory/pro-fibrotic component that leads to injury that cannot be addressed by revascularisation alone. Pathology specimens of patients with renovascular disease have shown inflammatory infiltrates of macrophages that contribute to fibrosis [13]. As will be discussed below, it is now believed that an influx of inflammatory cells within the compromised kidney leads to secretion of pro-inflammatory and pro-fibrotic cytokines such as monocyte chemoattractant protein 1(MCP-1), tumour necrosis factor alpha (TNF ∝), interferon gamma (IFN- γ) and interleukin 6 (IL-6), which contribute to the insult [14]. This suggests that although significant stenosis causes injury by reduced blood flow and increased activation of the plasma renin system, activation of the inflammatory and pro-fibrotic pathways contribute significantly to progressive renal impairment, which cannot be addressed by revascularisation alone.

### 3.2. Pathophysiology of Ischaemic Nephropathy

The initial reduction in renal blood flow associated with renal artery stenosis (RAS) leads to activation of the renin-angiotension-aldosterone system (RAAS), a physiological response that leads to increase in MAP in order to restore perfusion pressure in the kidney beyond the stenosis. Some studies have implicated that this initial increase in renin and angiotension II is part of a self-perpetuating mechanism that maintains systemic vasoconstriction [15] since rising levels of angiotension II lead to increase in levels of prostaglandin F2α-isoprostanes, which leads to a decrease in renal blood flow, sodium retention and vasoconstriction. Activation of the RAAS system is pro-inflammatory, but it also contributes to renal injury by enhancing expression of profibrotic cytokines such as transforming growth factor beta (TGF-β), leading to tubulointerstitial fibrosis [16]. The activation of the RAAS itself can generate reactive oxygen species and in turn increase oxidative stress. (Figure 1)

Much of our recent understanding has been derived from pig models. Following experimental induction of RAS, levels of prostaglandin G2 α-isoprostrane (a systemic marker of oxidative stress) along with other markers of oxidative stress are markedly increased and correlate well with both renin activity and arterial pressure. Even when renin activity returns to normal, arterial pressure remains elevated along with isoprostrane levels, possibly due to a prolonged effect of angiotension II within the ischaemic tissue, demonstrating a correlation between renovascular hypertension and oxidative stress [17]. As stenotic lesions progress leading to more hypoperfusion of the renal cortex, and hence more oxidative stress, this in turn leads to inflammation and tubular injury in the affected kidney, leading to irreversible injury, fibrosis and ultimately progression of renal impairment to ESKD [18]. This persistent inflammation and related oxidative stress likely explain the failures of revascularisation to have a marked effect on estimated Glomerular Filtration Rate (eGFR) and blood pressure in the majority of patients with atherosclerotic RAS. Human studies have shown improved renal blood flow and cortical prefusion post successful revascularisation, but, despite this, markers of renal inflammation such as neutrophil gelatinase-associated lipocalin (NGAL), MCP-1 and TNF-α remain elevated and are not altered by restoration of blood flow or tissue oxygenation [19]. MCP-1 in particular has a key role as a chemokine for macrophage infiltration [20]. TGF-α expression as a marker of macrophage infiltration is much higher in kidneys affected by atheromatous renal artery disease in comparison to the contralateral unaffected kidney [21]. Hence, although RAS does lead to renal hypoxia, revascularisation alone fails to alter the inflammatory and oxidative stress components of the pathophysiology, and they play a key role in the progression of injury.

Chronic hypoxia ultimately leads to apoptosis of epithelial cells, which in turn further activates inflammatory and fibrotic pathways that contribute to progression to ESKD [22]. Microvascular changes are also important, as shown in porcine models. Angiogenesis is an important compensatory mechanism to revascularize ischaemic tissue, but this is impaired by the chronic hypoxia of RAS. Hypoxia decreases VEGF stimulated phosphorylation of VEGF receptors (KDR: kinase domain receptor), which in turn leads to reduced expression of the KDR protein, which plays a vital role in the angiogenic response in tissue that is under threat [23]. In humans, expression of VEGF is seen to be low amongst patients with atheromatous renovascular disease in comparison to other causes of renal impairment such as diabetic nephropathy [24].

### 3.3. Markers of Oxidative Stress for Research Application

Oxidative stress develops due to an imbalance of exposure to free radicals mainly derived from oxygen and of antioxidant defences such as gluthathione and antioxidant enzymes such as superoxide dismutase. Several markers of oxidative damage exist (Table 2), including:
(1)Isoprostanes, which are prostaglandin-like substances produced independently of the cyclooxygenase (COX) enzymes but mainly via oxygen radical-induced peroxidation of arachidonic acid [25]. Studies in pig models with unilateral RAS show that in early renovascular hypertension, an increase in plasma renin activity and arterial pressure is associated with rising markers of oxidative stress—predominantly isoprostanes. The progressive increase in the levels of isoprostanes mirrors the increase in mean arterial pressure (MAP), even when plasma renin returns to baseline. This supports the link between activation of the RAAS and the oxidative stress pathway, which plays a key role in the renal injury seen in renal artery disease [17].(2)Thiobarbituric acid reactive substances (TBARS) have been widely used as measures of lipid peroxidation in biological fluids. They are considered a reliable indicator of oxidative stress [26]. Animal studies have shown high levels of TBARS in models of renovascular disease such as that of the ‘‘2 kidneys 1 clip’’ model of hypertension; they have been shown to correlate with mean arterial pressure and oxidative stress in the ischaemic kidney [27].(3)Neutrophil gelatinase-associated lipocalin (NGAL) is an acute-phase protein expressed by activated neutrophils and tubular epithelial cells when exposed to both inflammatory and ischaemic conditions [28]. It is commonly used as a predictive biomarker for acute kidney injury [29]. In ischaemic mouse models, NGAL mRNA is initially upregulated in the post stenotic kidney and this is closely followed by increased urinary NGAL within 2 h of the ischaemic insult [30]. In a human study, NGAL levels were elevated both systemically and in affected RAS kidney renal veins in patients with unilateral RAS compared to those with either normal blood pressure or essential hypertension. This supports a role of persistent inflammation being important in perpetuating renal injury beyond a stenotic lesion in renovascular hypertension [31].(4)Insulin-like growth factor binding protein 7 (IGFBP-7) and tissue inhibitor of metalloproteinases-2 (TIMP-2) are both biomarkers of cell cycle arrest, which are often elevated in ischaemic conditions. When exposed to oxidative stress, renal tubular cells produce and release IGFBP 7 and TIMP-2, which stimulate the expression of p53, p21 and p27, all of which affect the cell cycle promotion resulting in transient arrest of the cell cycle [32]. Sustained cell cycle arrest ultimately leads to fibrosis.

Both of these factors have been validated as urinary biomarkers for acute kidney injury in intensive care settings [33].

Renal vein levels of NGAL, IGFBP7 and TIMP-2 are elevated in those with stenotic vascular lesions and decrease immediately post revascularisation; however, they are seen to return to baseline 3 months after revascularisation. As stated previously, the processes of inflammation and injury in the RAS kidney persist long after renal revascularization [34].

### 3.4. Imaging Biomarkers of Ischaemia and Potential Oxidative Damage

Blood-oxygen-level-dependent (BOLD) magnetic resonance imaging is a functional MRI technique that relies on regional differences in blood flow to reflect tissue oxygenation. Due to the countercurrent arrangement of intra-renal vessels and renal tubules to facilitate water reabsorption and the formation of concentrated urine, there exists a gradient of PO_2_ between the blood vessels of the medulla and cortex. Oxyhemoglobin is diamagnetic and deoxyhemoglobin is paramagnetic, which leads to local dephasing of protons and an increase in “relaxivity” during MRI that is known as R2*. The amount of deoxyhemoglobin functions as a contrast agent. The cortex is identifiable due to it having lower R2* levels, with increasing levels seen in deeper medullary regions [35]. MRI studies have indicated that alteration in oxygen consumption in regions of interest can be detected more effectively by higher magnetic field strength [36].

Renal hypoxia can be an important contributor in many cases of all-cause progressive CKD, as hypoxia acts as a stimulus for the inflammatory pathway and eventually to fibrosis [37]. Previous studies using BOLD MRI have shown that cortical R2* levels are higher in those with all-cause CKD in comparison to healthy controls and that R2* correlates positively with rate of decline of eGFR, confirming that oxygenation is reduced in those with CKD [38].

BOLD MRI has also been used to highlight the decreased cortical oxygenation in those with ARVD, and this has potential to a be a valuable tool to highlight kidneys under ischaemic stress. A study that included 30 patients with atheromatous RAS, 17 of whom were considered severe as defined by ultrasound velocities more than 384 cm/sec, illustrated that cortical R2* levels were elevated in those with severe ARAS in comparison to those with either essential hypertension or even moderate ARAS. This indicates that kidneys with more severe RAS lesions are unable to adapt to the reduced renal blood flow as shown by overt cortical hypoxia measured by BOLD-MRI. In contrast, moderate RAS was associated with preserved cortical and medullary oxygenation, indicating an adaptation to reduced renal blood flow that may explain how renal function can remain stable throughout treatment with anti-hypertensive therapy such as renin-aldosterone-angiotensin inhibition in patients with ARVD [35].

BOLD MRI may also help in the management of those with atheromatous RAS by identifying kidneys at risk of further injury. Demonstration of cortical hypoxia may be used to discriminate between acute and chronic lesions and thus identify those most likely to benefit from revascularisation, this being more applicable to the acute situation. One study using 3.0 Tesla MRI examined patients with RAS who would have met the entry criteria for the CORAL trial and who were receiving renin-angiotensin-aldosterone inhibitors. R2 levels in those with a stenotic lesion sufficient to cause an elevated level of renin were similar to those with essential hypertension, indicating a protective effect of RAAS inhibition in maintaining oxygenation. However, the medullary R2* response to furosemide administration was reduced in those with a stenotic lesion compared to the contralateral kidney, which increased its metabolic activity as compensation for the stenotic kidney [39]. Such compensation may indicate the difference between an acute and chronic lesion, and it may explain the lack of additional benefit of revascularisation over medical therapy, as seen in the large revascularisation trials. Those trials focused on the reversing the haemodynamic insult, whereas now there is a better understanding of the ongoing inflammatory insult, which requires a multi-faceted approach.

Atheromatous RAS leads to reduced oxygenation and reduced GFR. Studies have shown a correlation between changes in oxygenation as seen on BOLD imaging and the increased inflammatory biomarkers that have been described previously. Reduced blood flow in RAS kidneys with resulting tissue hypoxia is associated with higher levels of NGAL and MCP-1. In the study of 30 ARVD patients by Hermann et al. that included (20 undergoing medical therapy and 10 renal artery stenting), levels of both NGAL and MCP-1 at baseline correlated directly with all levels of hypoxia in the stenotic kidney. These levels did not return to normal with either intensive medical therapy or revascularisation. Despite restoration of blood flow, there was a strong relationship between the biomarkers and BOLD assessed residual hypoxia at 3 months post revascularisation [21]. This supports the fact that ongoing inflammation persists after revascularisation in atherosclerotic RAS, and this highlights the need for additional medical interventions to target the inflammatory and profibrotic processes within the renal parenchyma.

### 3.5. Potential Therapeutic Interventions to Address the Inflammatory and Oxidative Damage in Ischaemic Nephropathy

Large randomised control trials such as ASTRAL [9] and CORAL [10] have effectively shown that restoring renal blood flow in unselected patients with RAS does not confer an additional benefit to optimal medical therapy. However, as discussed previously ARVD is a complex disease involving multiple pathophysiological pathways, including ischaemia, inflammation, fibrosis and cell death, with other contributing risk factors such as dyslipidaemia. As such, these pathways offer potential for alternative therapeutic interventions.

#### 3.5.1. Vascular Endothelial Growth Factor (VEGF)

VEGF is a pro-angiogenic factor that promotes cell division, survival and mobilisation of endothelial progenitor cells and also vascular proliferation [40]. It also plays a key role in angiogenesis, which is important in the renal tissue’s response to ischaemia. Reduced VEGF activity has been shown to play a role in a variety of renal conditions, including hypertension, chronic kidney disease [41] and (especially) renovascular disease [42].

In animal models of non-renovascular CKD, VEGF treatment has been shown to preserve the microvascular architecture, and it stimulates endothelial progenitor cells, which in turn leads to more stable renal function and slowing of disease progression [43,44]. Specific to atherosclerotic RAS, it has been shown that the intra-renal administration of recombinant human VEGF through the renal artery distal to a stenotic lesion in swine models has a renoprotective effect with improved intra-renal vasculature, less progressive renal dysfunction and less fibrosis, even when administered at an advanced stage of the disease process [45]. This is quite a promising therapy, and there are multiple ongoing studies investigating how this might be best delivered to those with ARVD and CKD using advances in drug delivery technology [46].

#### 3.5.2. Mesenchymal Stem Cells

Mesenchymal stem cells (MSC) are multipotent adult cells that have the ability to differentiate into several cell types. They have been shown to be effective in several animal models of renal disease, with roles in repair and reducing renal injury demonstrated in CKD [47]. In swine models of RAS, intra-renal delivery of MSC has been shown to restore expression of angiogenic factors, to reduce vascular remodelling and to reduce tissue injury within the post-stenotic kidney. This model has highlighted that increased markers of oxidative stress are evident in the stenotic kidney, including increased TNF-∝ and MCP-1 expression in interstitial cells with increased IL-10 in the renal tubules, and the administration of MDC attenuated the release of these and other inflammatory markers from the post-stenotic kidney [48]. These studies suggest that MSC may play a role in modulating pathophysiological pathways, including inflammation and fibrosis, to improve renal function.

A phase 1 trial has investigated the potential of autologous adipose-derived MSCs infused through the renal artery in 21 patients with ischaemic nephropathy. Three different dose levels were used (each in 7 patients) and were compared to 18 medically managed patients. The MSC treatment resulted in a significant drop in hypoxia and inflammatory cytokines, with a fall in blood pressure and an improvement in GFR. The benefit was greatest in those receiving the highest MSC dose [49].

There is currently ongoing trial determining whether an infusion of participants’ own MSCs, harvested from a fat pad biopsy and given prior to angioplasty, enhances renal blood flow and renal function post revascularization [50].

#### 3.5.3. Targeting Inflammation

As discussed earlier, inflammation plays a key role in the pathophysiology of ischaemic nephropathy with elevation of biomarkers such as NGAL, TNF-∝ and infiltration of the renal parenchyma by inflammatory cells such as macrophages. The predominant aetiology in ischaemic nephropathy is atherosclerosis, which is characterised by a chronic low-grade inflammatory state that can precipitate inflammatory infiltration of the renal parenchyma [51].

Previous studies in other causes of chronic renal disease such as glomerulonephritis have shown benefit from using immunodepleting therapy, such as mycophenolate mofetil to reduce the inflammatory infiltrate of macrophages and lymphocytes and in turn reduce sclerosis and proteinuria. Interestingly, the benefits of such therapy has also been shown in non-immune models of disease, with a reduction of progression of renal function and histological changes [52]. Such anti-inflammatory therapy has not been tested in ischaemic nephropathy, but this therapy might have some potential.

#### 3.5.4. Anti-Fibrotic Therapy

Chronic inflammation ultimately leads to renal fibrosis, which is a common final pathway in progression of CKD to ESKD. In ischaemic nephropathy, the activation of the RAAS and increase in oxidative stress and inflammation all lead to the development of renal fibrosis. Given its key role, therapeutic interventions to prevent or delay the progression of fibrosis have the potential to have significant impact [53].

TGF-β is a key mediator of kidney fibrosis and as such is a potential target for anti-fibrotic strategies [54]. Fresolimumab, a monoclonal antibody against TGF-β, has been trialled in randomised studies of 36 patients with steroid resistant focal segmental glomerulosclerosis (FSGS). Although the study was underpowered to meet primary or secondary endpoints, fresolimumab was well tolerated and relatively safe opening the opportunity for evaluation in other renal conditions [55].

Pirfernidone is an orally active anti-fibrotic agent that has been used in multiple conditions including pulmonary fibrosis. In animal- and cell-based models, it has been shown to be an effective anti-fibrotic treatment due to its modulating effect on cytokines and reducing TGF-β [56]. Its effective renal anti-fibrotic effect has been shown across a number of renal conditions including obstruction [57], post sub-total nephrectomy [58] and diabetic nephropathy [59]. In these models, the use of pirfenidone reduced proteinuria, reduced the rate of decline of GFR, decreased interstitial fibrosis and decreased macrophage infiltration. In a randomised, double-blind controlled study of 77 participants with diabetic nephropathy manifest by albumuria and reduced eGFR, treatment with pirfenidone was associated with increased eGFR after one year. The eGFR increased by a mean of 3.3 mL +/−8.5 mL/min/1.73 m^2^ in the treated patients compared to a decrease of 2.2 mL +/−4.8 mL/min/1.73 m^2^ (*p* = 0.026) in placebo treated patients [60].

Although pirfenidone has not been assessed in clinical trials of ischaemic nephropathy, in an animal model of renal ischaemia prophylactic pirfenidone conferred protection against the resulting AKI. It was demonstrated that pirfenidone had an antioxidant effect that was possibly mediated by improvement in nitrate generation after the ischaemic insult. There was also a reduction in the urinary albumuria: creatinine ratio (ACR) in those treated with pirfenidone [61]. The possibility of using pirfenidone for the prophylactic prevention of AKI in patients exposed to potential ischaemic insults, such as transplantation, cardiovascular surgery or RAS, is therefore raised.

## 4. Conclusions

Ischaemic nephropathy is far more complex than simply being due to a haemodynamic issue; multiple pathophysiological pathways, with prominence of inflammation and oxidative stress, interact to ultimately lead to fibrosis, progressive renal impairment and, if left unchecked, to ESKD. To date, therapeutic interventions have focused on reducing risk factors with RAAS inhibition, lipid lowering agents and hypoglycaemic agents in those with diabetes long with renal revascularisation. Our current approach thus overlooks the complex interactions at play distal to the stenotic lesion in the renal parenchyma in which oxidative stress and chronic inflammation often persist.

The pre-clinical and human evidence now suggest that where a clinical indication for revascularization is present, the focus should be on renoprotective therapies that combine this with novel therapeutic interventions that directly target the renal tissue response to ischaemia, reduce the inflammatory infiltrate and prevent or reduce the fibrosis.

## Figures and Tables

**Figure 1 antioxidants-10-00845-f001:**
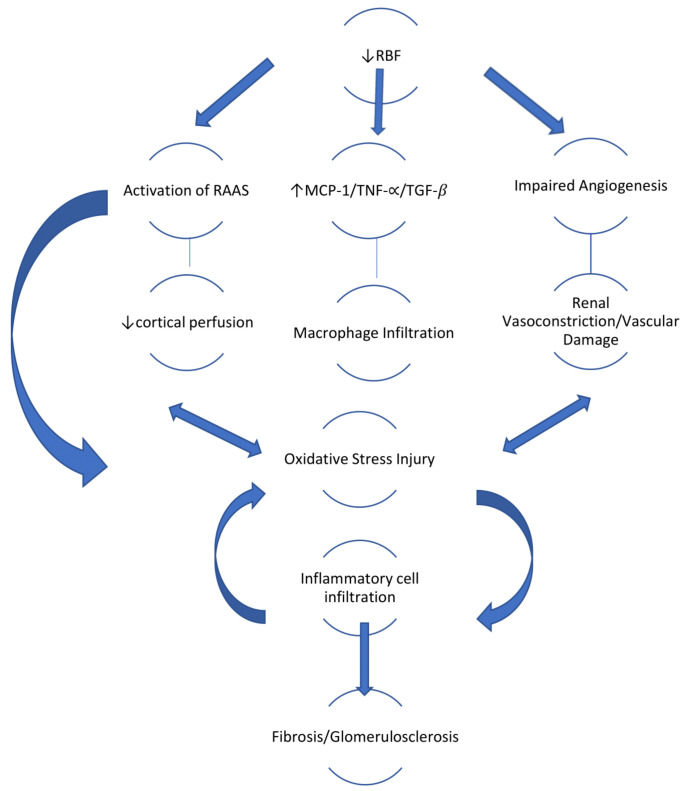
Role of oxidative stress in the pathophysiology of ischaemic nephropathy.

**Table 1 antioxidants-10-00845-t001:** Randomised control trials of renal revascularisation versus medical therapy.

Author	Year	Cohort	Follow-Up (Months)	End Points	Key Points	Limitations
Bax et al [8] **STAR**	2009	140 patients with RAS ≥50% CrCl <80ml/min BP < 140/90 Randomised to medical therapy (76) vs medical therapy + PTRAS (64)	24	>20% decrease in CrCl from baseline: 22% in medical vs 16% in medical + PTRAS Complications Mortality 8% in both groups	PTRAS did not confer a benefit but led to significant procedure related complications such as: 2/62 (3%) periprocedural deaths 1 death secondary to infected haematoma 1 patient with ESKD necessitating RRT	Significant number of lesions <50% at inclusion (19%)
Wheatley et al [9] **ASTRAL**	2009	806 patients with “substantial” RAS Uncertain benefit from revascularisation Randomised to medical therapy (403) vs medical therapy + PTRA/PTRAS (403)	33.6	The rate of progression of renal impairment (as shown by the slope of the reciprocal of the serum creatinine level favoured revascularization. BP: no significant difference in systolic BP ESKD: 8% in both groups CVE: 35% in medical group bs 36% Mortality: 26% in both groups	No meaningful benefit from revascularisation in patients with ARAS. Significant risk associated with revascularisation: 2 deaths 3 amputations	Population studied- only if own physician was uncertain about the benefit of revascularisation. 20% had stenosis <50%
Cooper et al [10] **CORAL**	2014	947 patients with RAS ≥60% Randomised to medical therapy (480) vs. medical therapy + PTRAS/PTRA (467)	43	Composite end point of death from CV or renal causes, MI, CVA, hospitalisation for HF, progressive renal impairment or need for RRT: 35.8% in medical group vs 35.1% Mortality: 16.1% in medical group vs 13.7% ESKD: 1.7% in medical group vs 3.5%	Renal-artery stenting did not confer a significant benefit with respect to the prevention of clinical events when added to comprehensive, multifactorial medical therapy in people with atherosclerotic renal-artery stenosis and hypertension or chronic kidney disease	Patients could be enrolled with lesion <60% Broad inclusion criteria. Higher baseline eGFR than ASTRAL (58ml/min vs 40ml/min).

**STAR**: The benefit of stent placement and blood pressure and lipid lowering for prevention of progression of renal dysfunction caused by atherosclerotic ostial stenosis of renal artery. **CORAL**: Cardiovascular Outcome in Renal Atherosclerotic Lesions. **ASTRAL**: Angioplasty and stenting for renal artery lesions. PTRAS: percutaneous transluminal renal angioplasty with stenting. CrCL: Creatinine Clearance. RRT: renal replacement therapy. ARAS: atherosclerotic renal artery stenosis. ESKD: end-stage kidney disease. CV: cardiovascular HF: heart failure.

**Table 2 antioxidants-10-00845-t002:** Trials of biomarkers for oxidative stress.

Author	Year	Cohort	Biomarker	Findings
Lerman et al. [17]	2001	Fourteen pigs with induced RAS (7) vs. sham procedure (7)	PGF_2α_-isoprostanes	RAS = ↑ PGF_2α_-isoprostanes, correlated with arterial pressure
Oliveria-Sales et al. [27]	2008	Fifty-eight rats 6 weeks post renal surgery using Goldblatt hypertension model 2K 1C	Thiobarbituric acid-reactive substances (TBARS)	Hypertensive group showed a significant increase in oxidative stress as compared to the control group (CT, 1.6 ± 0.3 nmol/mL vs. 2K-1C, 2.23 ± 0.4 nmol/mL)
Eirin et al. [31]	2012	Patients with EH (*n* = 22) or unilateral renal artery stenosis (*n* = 22) and normotensive controls (*n* = 22). Urine samples were also collected from 16 consenting healthy age-matched potential kidney donors	Plasma/urinary neutrophil gelatinase-associated lipocalin (NGAL) Renal inflammatory markers (IF-γ, TNF α)	↑ NGAL levels in both the systemic circulation and the stenotic kidney vein of RVH compared with EH patients. Renal vein levels of IF-γ and TNF-α were higher only in the stenotic kidney vein compared with normal and EH (*p* < 0.05).
Wang et al. [34]	2016	ARAS patients (*n* = 12) with clinical indications for renal revascularization vs. essential hypertension (EH) (*n* = 12)	NGAL, MCP-1, IL-10, TNF α, KIM-1, IGFBP7, TIMP-2	Baseline renal venous levels of NGAL, IGFBP7, TIMP-2, MCP-1 and TNF-α ↑ in STKs vs. EH IGFBP7/TIMP-2/MCP-1 ↓ immediately post revascularisation NGAL ↔ All except NGAL returned to baseline at 3/12 Biomarkers remained higher than those observed in EH

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
