# Peer review of "Inflammation and Oxidative Damage in Ischaemic Renal Disease"

_antioxidants, 2021, doi:10.3390/antiox10060845_

Round 1

Reviewer 1 Report

General comments

This is an interesting thematic review addressing the impact of inflammation and oxidative stress in IRD. Overall, the topics are well chosen, and the review is well written and comprehensive. Nevertheless, the descriptive point by point form of writing is not very appealing or easy to follow for the reader. The manuscript would greatly benefit from the inclusion of schemes/figures that could illustrate the actors in the interplay between inflammation oxidative stress and injury at cell and tissue level. Also, the readers would benefit from tables that could summarize all the information on the clinical studies characterization and major findings as described in the manuscripts description specifically on the section dedicated to biomarkers of ischaemia and oxidative damage.

Detailed comments

Since this is a review manuscript the relevance of a correct description of citations is of the most importance for the readers and the journal. The list of references need a detailed revision since it currently includes a series of typos and in some cases are not complete lacking the volume/number/page, incomplete (e.g.27, 63,66..)

Author Response

Many thanks for your kind remarks. I have included a schematic of the various oxidative stress pathways involved in the pathophysiology of ischaemic nephropathy in the relevant section. 

I have also included 2 Tables. Table 1 highlights the major clinical trials to date in terms of revascularisation compared to medical therapy of those with atherosclerotic renal artery stenosis. 

Table 2 highlights the trials involving biomarkers of oxidative stress in ischaemic nephropathy. I hope these tables provide the concise relevant information. 

My apologies with regards the references I believe these have now been appropriately addressed.

Reviewer 2 Report

The article is well set up, clear and complete.
Perhaps there are too many abbreviations that make the reading not exactly fluent.

Author Response

Many thanks for your remarks. I have now ensure all abbreviations have been explained/expanded.

Reviewer 3 Report

The review article entitled "Inflammation and Oxidative Damage in Ischaemic Renal Disease" covers a considerable amount of information on renal ischemia-related inflammation and oxidative stress. The following points should be considered before accepting the article:

The authors should provide a graphical summary for the review article.

Line 84: period is missing.

Line 85: MAP full-form is not described anywhere earlier in the text.

Line 95: replace "TNK α" with "TNF-α" also "IL-y" by "IFN-y."

Line 103: RAS full-form is not described anywhere earlier in the text.

Line 107: replace 'as rising' with 'As rising.' 

Rephrase paragraph (lines 148-152) for the sake of readers' understanding.

Line 203 & 205: What does the asterisk(*) on R mean? It is not explained anywhere in the text.

Page # 5: Are R2 and R2* the same? If yes, why written differently, and if not, how are they different?

Line 231: The authors should throw some insight on CORAL. The review articles are meant to provide an insight to the readers, even to someone new to the field. 

Rephrase paragraph (lines 242-244) " Atheromatous RAS leads to ------- described previously." for readers' understanding.

Line 259, 274 & 335: correction needed reference style. 

Line 306: Delete 'on.'

Author Response

Many thanks for your remarks and the appropriate edits have been made.

R2 and R2* are the same and is the symbol used to  abbreviate for relaxivity on MRI as explained on line 245. My apologies for any confusion caused.

Round 2

Reviewer 3 Report

All the points raised in the previous version of the manuscript have been addressed.